# NaViL: Rethinking Scaling Properties of Native Multimodal Large Language Models under Data Constraints

**Changyao Tian**[2,1*†]  **Hao Li**[1*†]  **Gen Luo**[1*]  **Xizhou Zhu**[3,1*]  **Weijie Su**[1]
**Hanming Deng**[4]  **Jinguo Zhu**[1]  **Jie Shao**[5,1†]  **Ziran Zhu**[4]  **Yunpeng Liu**[4]
**Lewei Lu**[4]  **Wenhai Wang**[2,1]  **Hongsheng Li**[2]  **Jifeng Dai**[3,1⊠]

[1] Shanghai AI Laboratory    [2] The Chinese University of Hong Kong
[3] Tsinghua University    [4] Sensetime Research    [5] Nanjing University
Code: `https://github.com/OpenGVLab/NaViL`

## Abstract

Compositional training has been the de-facto paradigm in existing Multimodal Large Language Models (MLLMs), where pre-trained visual encoders are connected with pre-trained LLMs through continuous multimodal pre-training. However, the multimodal scaling property of this paradigm remains difficult to explore due to the separated training. In this paper, we focus on the native training of MLLMs in an end-to-end manner and systematically study its design space and scaling property under a practical setting, *i.e.,* data constraint. Through careful study of various choices in MLLM, we obtain the optimal meta-architecture that best balances performance and training cost. After that, we further explore the scaling properties of the native MLLM and indicate the positively correlated scaling relationship between visual encoders and LLMs. Based on these findings, we propose a native MLLM called NaViL, combined with a simple and cost-effective recipe. Experimental results on 14 multimodal benchmarks confirm the competitive performance of NaViL against existing MLLMs. Besides that, our findings and results provide in-depth insights for the future study of native MLLMs.

## 1 Introduction

Multimodal Large Language Models (MLLMs) have demonstrated remarkable progress in computer vision [12, 43, 63, 50, 54], continuously breaking through the upper limits of various multimodal tasks [47, 68, 38, 45]. The great success of MLLM is inseparable from its compositional training paradigm, which independently pre-trains visual encoders [28] and LLMs [61], and then integrates them through additional multimodal training. Due to the engineering simplicity and effectiveness, this paradigm has dominated MLLM area over the past few years. However, the shortcomings of compositional training have been gradually recognized by the community recently, *e.g.*, unclear multimodal scaling property [19, 56].

Therefore, increasing attention has been directed toward the development of more native MLLMs. As illustrated in Fig. 1, native MLLMs aim to jointly optimize both visual and language spaces in an end-to-end manner, thereby maximizing vision-language alignment. Compared to the compositional paradigm, existing native MLLM methods demonstrate a promising scaling law and a significantly simplified training process [9, 56]. Despite these advancements, the primary benefits of native MLLMs are often evaluated under the assumption of infinite training resources, overlooking the

---

\* Equal contribution. ⊠ Corresponding to Jifeng Dai <daijifeng@tsinghua.edu.cn>.
† Work was done when Changyao Tian, Hao Li, and Jie Shao were interns at Shanghai AI Laboratory.

39th Conference on Neural Information Processing Systems (NeurIPS 2025).

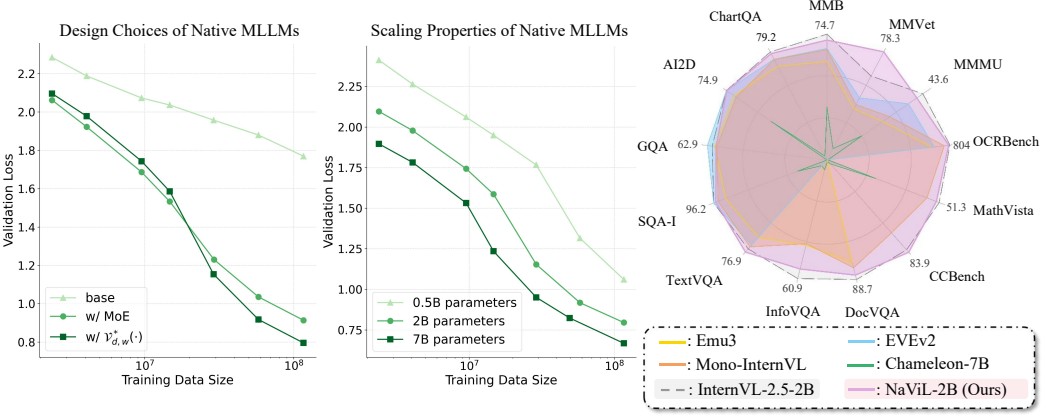

Figure 1: **Comparison of design choices, scaling properties, and performance of our native MLLMs**. We systematically investigate the designs and the scaling properties of native MLLMs under data constraints and yield valuable findings for building native MLLMs. After adopting these findings, our native MLLMs achieve competitive performance with top-tier MLLMs. $\mathcal{V}_{d,w}^*(\cdot)$ denotes the visual encoder with optimal parameter size.

substantial challenges posed by limited data and large-scale training. Consequently, a critical practical question remains: whether and how native MLLMs can feasibly achieve or even surpass the performance upper bound of top-tier MLLMs at an acceptable cost.

To answer this question, in this paper, we aim to systematically investigate the designs and the scaling properties of native MLLMs under data constraint. Specifically, we first explore the choices of key components in the native architecture including the mixture-of-experts, the visual encoder and the initialization of the LLM. Our findings can be summarized in two folds. Firstly, an appropriate pre-training initialization (*e.g.,* the base LLM) of the LLM greatly benefits the training convergence on multimodal data. Secondly, combining visual encoder architectures and MoEs results in obvious gains against the vanilla decoder-only LLM. Following these findings, we build a meta architecture that optimally balances performance and training cost.

Based on the optimal meta architecture, we further explore the scaling properties of the visual encoder, the LLM and the entire native MLLM. Specifically, we first scale up the LLM and the visual encoder independently and observe different scaling properties: while scaling LLM exhibits similar patterns as the conventional language scaling laws, scaling visual encoder shows an upper bound in return due to the limitation of the LLM's capacity, suggesting that the optimal encoder size varies with the LLM size. Further analysis reveals that the optimal encoder size increases approximately proportionally with the LLM size in log scale. This observation yields a different guidance against compositional paradigm, which employs a visual encoder of one size across all LLM scales.

Based on above principles, we propose a native MLLM called NaViL, combined with a simple and cost-effective recipe. To validate our approach, we conduct extensive experiments across diverse benchmarks to evaluate its multimodal capabilities including image captioning [10, 67, 2], optical character recognition (OCR) [57, 17, 39], *etc*. Experimental results reveal that with ~600M pre-training image-text pairs, NaViL achieves competitive performance compared to current top-tier compositional MLLMs, highlighting the great practicality and capabilities of NaViL. In summary, our contributions are as follows:

- We systematically explore the design space and the optimal choice in native MLLMs under data constraint, including the LLM initialization, the visual encoder and the MoEs, and draw three critical findings that greatly benefit the training of native MLLMs.

- Based on above findings, we construct a novel native MLLM called NaViL. In NaViL, we explore the scaling properties of the visual encoder and the LLM and indicate their positively correlated scaling relationship.

- We conduct large-scale pre-training and fine-tuning experiments on NaViL. Experimental results show that NaViL can achieve top-tier performance with nearly 600M pre-training data. Our findings and results will encourage future work for native MLLMs in the community.

## 2 Related Work

**Multimodal Large Language Models.** Recent years have witnessed the significant progresses of Multimodal Large Language Models (MLLMs) [44, 36, 35, 63, 12], which have dominated various downstream tasks [24, 26, 57, 30]. Starting from LLaVA [36], most existing MLLMs adopt the compositional paradigm, which connects the pre-trained visual encoder [53] and LLM [3] through a projector and finetune them on for alignment. Then, the whole structure will be further fine-tuned on multimodal data for alignment. Based on this paradigm, existing works mainly focus on the improvement of visual encoders [63, 64, 44] and the design of connectors [33, 36]. Despite the progress, such paradigm struggles to explore the joint scaling properties of vision and language. Their potential limitations in training pipeline [56] and vision-language alignment [19] are also gradually recognized by the community.

**Native Multimodal Large Language Models.** To overcome the limitations of compositional paradigm, native MLLMs have emerged as another candidate solution [20, 19, 43, 32, 62, 56, 9]. Compared to compositional paradigm, native MLLMs aim to pre-train both vision and language parameters in an end-to-end manner, thus achieving better alignment. The most representative methodology [56, 9] is to directly pre-train the LLM from scratch on large-scale multimodal corpora, which typically requires expensive training costs. To address this issue, recent attempt initialize the LLM with a pre-trained checkpoint to facilitate training convergence [20, 19, 43, 32, 62]. Nevertheless, current research still lacks systematic investigation into the architectural design and scaling characteristics of native MLLMs, limiting their performance.

## 3 Visual Design Principles for native-MLLM

### 3.1 Problem Setup

We define native MLLMs as models that jointly optimize vision and language capabilities in an end-to-end manner. Dispite recent progress that shows promising scaling law and potential better performance compard with their compositional counterparts, how to build competitive native MLLMs compare to the state-of-the-art MLLMs with a practical data scale remains underexplored. In particular, there are two problems requiring to be investigated:

- (Sec. 3.2) How to choose the optimal architectures of the visual and linguistic components?
- (Sec. 3.3) How to optimally scale up the visual and linguistic components?

**Meta Architecture**. To study these two questions, we first define a general meta architecture of native MLLMs consisting of a visual encoder, an LLM, and a mixture-of-expert architecture injected to the LLM. The visual encoder $\mathcal{V}$ consists of a series of transformer layers and can be defined as

$$\mathcal{V}_{d,w}(I) = \mathcal{C} \odot \mathcal{F}_d^w \odot \cdots \odot \mathcal{F}_2^w \odot \mathcal{F}_1^w \odot \mathcal{P}(I) = \mathcal{C} \bigodot_{i=1\ldots d} \mathcal{F}_i^w \odot \mathcal{P}(I), \tag{1}$$

where $\mathcal{F}_i^w$ denotes the $i$-th transformer layer (out of $d$ layers) with hidden dimension $w$, $\mathcal{P}$ denotes the Patch Embedding Layer, $I \in \mathbb{R}^{H \times W \times 3}$ denotes the input image. Note that the visual encoder degenerate to a simple patch embedding layer when $d = 0$. For simplicity, we use the same architectures as the LLM for the visual encoder layers $\mathcal{F}$ but with bi-directional attention and vary the hyperparameters $d$ and $w$. Here $\mathcal{C}$ is the connector which downsamples the encoded image embeddings through pixel shuffle [15] and projects them to the LLM's feature space by a MLP.

**Experiment Settings**. All the models are trained on web-scale, noisy image-caption pair data [55] with Next-Token-Prediction (NTP) and an image captioning task. We use a held-out subset of the multimodal dataset to calculate the validation teacher-forcing loss for measuring and comparing different design choices. Models with LLM initializations are initialize from InternLM2-Base [8].

### 3.2 Exploring the Optimal Design of Architecture Components

In this section, we explore the design choices of three key components: 1) the initialization of the LLM; 2) the effectiveness of MoEs; 3) the optimal architecture of the visual encoder.

### 3.2.1 Initialization of LLM

A straightforward way to construct native MLLMs is to train all modalities from scratch with mixed corpora, as shown in prior work [56]. While this approach theoretically offers the highest performance

ceiling given ample data and computational resources, practical limitations such as data scarcity and large-scale optimization challenges hinder its feasibility. Alternatively, initializing the model from a pre-trained LLM effectively leverages linguistic prior knowledge, significantly reducing data and computational demands.

To evaluate the effectiveness of LLM initialization, we compare model performance in terms of loss and image captioning. As shown in Fig. 2 (left), the model trained from scratch performs significantly worse than the initialized model, requiring over 10x more data to reach comparable loss.

Further analysis of zero-shot image captioning (Fig. 2 (right)) reveals a substantial performance gap favoring the initialized model, even with significantly more data for the non-initialized model. This is likely due to the lower textual quality and diversity of multimodal training data compared to the LLM pre-training corpus, limiting the textual capability of models trained from scratch. These findings highlight the practical advantage of using LLM initialization in multimodal pre-training.

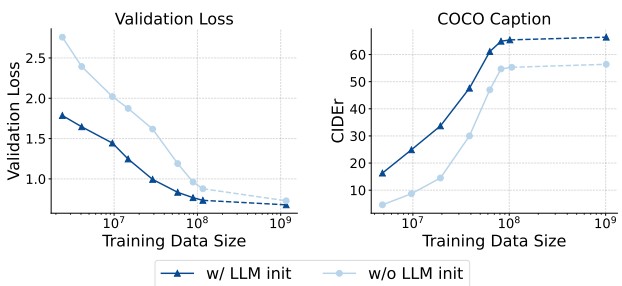

Figure 2: **Effectiveness of LLM initialization**. *Left*: The validation loss. The LLM initialized one converges much faster. *Right*: The zero-shot caption performance. Due to the lack of textual knowledge, the uninitialized model continues to lag behind.

> ***Observation 1:*** Initializing from pre-trained LLM greatly benefits the convergence on multimodal data, and in most cases delivers better performance even with a large amount of multimodal data.

### 3.2.2 Effectiveness of MoEs

Mixture-of-Experts (MoEs) are effective for handling heterogeneous data and are widely used in native MLLMs. We evaluate the MoE architecture within our meta architecture by comparing two configurations: one with a visual encoder and a vanilla LLM, and another with a visual encoder and an MoE-extended LLM. We follow Mono-InternVL [43] to adopt the modality-specific MoEs and training settings. However, we empirically found that using only the feed-forward network (FFN) expert would lead to a significant difference in feature scale between visual and language modalities. To mitigate this issue, we further introduced modality-specific attention experts, that is, using different projection layers (*i.e.* qkvo) in the self-attention layer to process visual and text features respectively, and then perform unified global attention calculation. Specifically, the output $x_{i,m}^l \in \mathbb{R}^d$ of the $i$-th token with modality $m \in \{\text{visual}, \text{linguistic}\}$ at the $l$-th layer of the MoE-extended LLM can be defined as

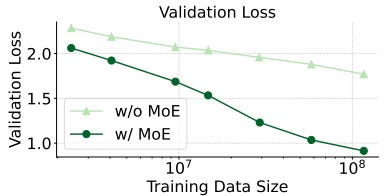

Figure 3: **The validation loss of adding MoE or not**. Using MoE extension will cause the loss to decrease more quickly.

$$
\begin{aligned}
x_{i,m}^{l'} &= x_{i,m}^{l-1} + \text{MHA-MMoE}(\text{RMSNorm}(x_{i,m}^{l-1})), \\
x_{i,m}^{l} &= x_{i,m}^{l'} + \text{FFN-MMoE}(\text{RMSNorm}(x_{i,m}^{l'})),
\end{aligned} \tag{2}
$$

where $\text{RMSNorm}(\cdot)$ is the layer normalization operation, and $\text{MHA-MMoE}(\cdot)$ and $\text{FFN-MMoE}(\cdot)$ are the modality-specific attention and FFN expert, respectively, formulated by

$$
\begin{aligned}
\text{MHA-MMoE}(x_{i,m}) &= (\text{softmax}(\frac{QK^T}{\sqrt{d}})V)W_O^m, \\
Q_{i,m} = x_{i,m}W_Q^m, K_{i,m} &= x_{i,m}W_K^m, V_{i,m} = x_{i,m}W_V^m, \\
\text{FFN-MMoE}(x_{i,m}) &= (\text{SiLU}(x_{i,m}W_{\text{gate}}^m) \odot x_{i,m}W_{\text{up}}^m)W_{\text{down}}^m.
\end{aligned} \tag{3}
$$

Here $W_Q^m, W_K^m, W_V^m, W_O^m$ and $W_{\text{gate}}^m, W_{\text{up}}^m, W_{\text{down}}^m$ are all modality-specific projection matrices, and $\text{SiLU}(\cdot)$ denotes the activation function, $\odot$ denotes the element-wise product operation. The number of activated experts is set to one to maintain consistent inference costs.

As shown in Fig. 3, the MoE architecture significantly accelerates model convergence compared to the vanilla LLM, achieving the same validation loss with only 1/10 of the data without increasing training or inference cost. This demonstrates that MoE enhances model capacity and effectively handles heterogeneous data, making it suitable for native MLLMs.

> ***Observation 2:*** MoEs significantly improve model performance without increasing the number of activated parameters.

### 3.2.3 Optimizing the Visual Encoder Architecture

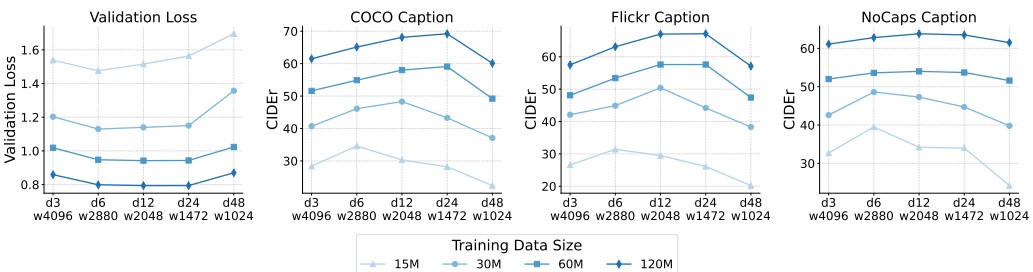

Figure 4: **The validation loss and zero-shot caption performance of different visual encoders**. The loss and performance only differ when the visual encoder is extremely wide or shallow.

The visual encoder precedes the LLM to perform preliminary extraction of visual information, converting raw pixels into semantic visual features aligned with the textual embedding space. Due to its bidirectional attention mechanism and the increased capacity introduced by additional parameters, the visual encoder has the potential to enhance the model's ability to represent visual information.

In this section, we investigate the optimal architecture of the visual encoder under a given parameter budget. The total parameter count $\mathcal{C}$ can be approximately calculated [29] as $\mathcal{N} = 12 \times d \times w^2$. Given a fixed $\mathcal{N}$, the structure of the visual encoder is mainly determined by its width $w$ and depth $d$.

**Depth** ($d$): Typically, deeper models can capture richer and more complex features, while also being more prone to gradient vanishing problems [58]. When it comes to MLLM, a visual encoder that is too shallow may not be able to extract enough high-level semantics, while a visual encoder that is too deep may cause low-level features to be lost, thus limiting the capture of fine-grained details.

**Width** ($w$): Compared to depth, width has relatively little impact on visual transformer performance [21], as long as it does not cause additional information bottlenecks. That is, it cannot be lower than the total number of channels within a single image patch. Under this premise, the width of the visual encoder does not have to be the same as the hidden size of the LLM.

We train various MLLMs with different $\mathcal{V}_{d,w}$ configurations (combinations of depth and width) while keeping the pre-trained LLM and visual encoder parameter count fixed at 600M. The depth $d$ ranges from $\{3, 6, 12, 24, 48\}$, and the width $w$ is adjusted as $\{4096, 2880, 2048, 1472, 1024\}$ to maintain a consistent parameter count. Fig. 4 shows the validation loss for different depth and width combinations as training data size varies. Models with extremely high or low depths perform worse than those with moderate configurations. Among reasonably configured models, shallower ones converge faster in the early phase (less than 30M data), but this advantage diminishes with more data. In zero-shot image captioning benchmarks, deeper visual encoders show slightly better performance, consistent with prior research on compute-optimal LLM architectures [29], which suggests a wide range of optimal width and depth combinations.

> ***Observation 3:*** Visual encoders achieve near-optimal performance across a wide range of depth and width configurations. Shallower encoders converge faster in early training, while deeper encoders perform slightly better with larger datasets.

### 3.3 Scaling Up Native MLLMs

In this section, we consider the scaling properties of our meta architecture. Specifically, we investigate: 1) the impact of scaling up the visual encoder and the LLM independently; 2) the optimal way of scaling the visual encoder and the LLM simultaneously. All models follow the optimal architecture discovered in Sec. 3.2, *i.e.*, with LLM initialization, MoEs, and optimal depth-to-width ratios of the visual encoders.

#### 3.3.1 Scaling up Visual Encoder and LLM Independently

We first investigate the scaling properties of the visual encoder and the LLM independently, *i.e.*, scaling up one component while keeping the other fixed. Specifically, we evaluate a series of LLMs with parameter sizes $\{0.5B, 1.8B, 7B\}$ and visual encoders with sizes $\{75M, 150M, 300M, 600M, 1.2B, 2.4B\}$.

**Scaling up LLMs.** The results are shown in Fig. 5. Scaling up the LLM parameters in native MLLMs exhibits a pattern consistent with the conventional LLM scaling law, where the loss decreases linearly as the parameter size increases exponentially.

**Scaling up Visual Encoder.** The results are shown in Fig. 6. In contrast to the LLM scaling law, increasing the visual encoder size does not consistently enhance multimodal performance. Instead, with a fixed LLM, the performance gains achieved by enlarging the visual encoder diminish progressively. Beyond a certain encoder size, further scaling results in only marginal loss reduction, indicating that the performance upper limit of the MLLM is constrained by the LLM's capacity.

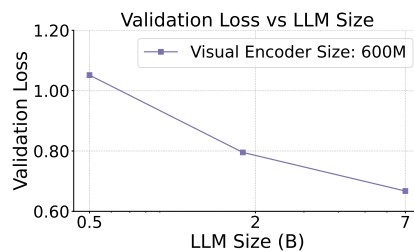

Figure 5: **The validation loss when scaling up LLMs**. With the same visual encoder (*i.e.* 600M), the validation loss decreases log-linearly with the LLM size.

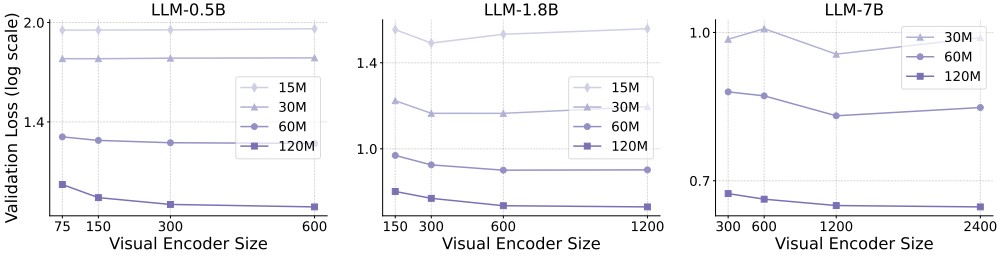

Figure 6: **The validation loss curves of different LLMs with different training data sizes**. As the training data size increases, the loss gap narrows to near zero when the visual encoder size reaches a certain threshold.

> ***Observation 4:*** Scaling the LLM consistently improves multimodal performance, following the typical LLM scaling law. However, increasing the visual encoder size shows diminishing returns, suggesting that the MLLM's performance is limited by the LLM's capacity.

#### 3.3.2 Scaling up Visual Encoder and LLM Together

The diminishing returns from increasing the visual encoder size suggest the existence of an optimal encoder size for a given LLM. We define this optimal size as the smallest encoder whose loss difference compared to an encoder twice its size is less than $\lambda = 1\%$ of the loss with the 75M encoder (the smallest used in our experiments). Fig. 7 shows the relationship between visual encoder size and LLM size.

The logarithm of the optimal visual encoder size scales linearly with the logarithm of the LLM size, indicating

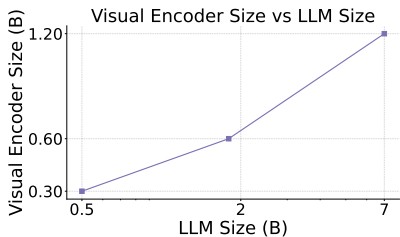

Figure 7: **Relationship of visual encoder size and LLM size**. The optimal visual encoder size increases log-linearly with the LLM size.

that both components should be scaled jointly for balanced performance. This highlights the suboptimality of compositional MLLMs, which typically use a fixed visual encoder size across varying LLM scales.

> ***Observation 5:*** The optimal size of the visual encoder scales proportionally with the LLM size in log scale, indicating that both components should be scaled jointly. This further implies that the pre-trained visual encoders using a single pre-trained visual encoder across a wide range of LLM scales like existing compositional MLLMs is suboptimal.

# 4 NaViL: A Novel Native MLLM with Strong Capabilities

## 4.1 Architecture

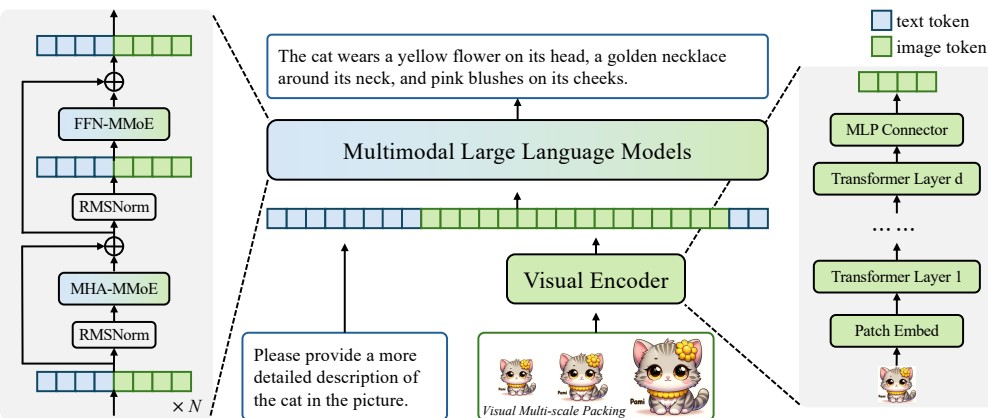

Figure 8: **Architecture of NaViL**. As a native MoE-extended MLLM, NaViL can be trained end-to-end and supports input images of any resolution.

Based on above studies, we construct NaViL with the optimal settings in Sec. 3.1. The architecture is shown in Fig. 8. NaViL inherently supports input images of any resolution. These images are first encoded into visual tokens by the visual encoder and the MLP projector, and then concatenated with the textual tokens to formulate the multimodal token sequence and fed into the LLM. Special tokens `<begin_of_image>` and `<end_of_image>` are inserted before and after each image token subsequence to indicate the beginning and end of the image, respectively. Special token `<end_of_line>` is inserted at the end of each row of image tokens to indicate the corresponding spatial position information.

**Visual Multi-scale Packing** is further introduced to improve the model performance during inference. Specifically, given an input image $I_0 \in \mathbb{R}^{H_0 \times W_0 \times 3}$ and downsampling rate $\tau$, a multi-scale image sequence $\{I_i \in \mathbb{R}^{H_i \times W_i \times 3}\}_{i=0}^n$ is obtained by continuously downsampling the original image (*i.e.* $H_i = \tau^i H_0, W_i = \tau^i W_0$) until its area is smaller than a given threshold. These images in the sequence are processed separately by the visual encoder. The obtained visual token embeddings $\{x_{i,v}\}_{i=0}^n$ are then concatenated and fed to the LLM. Special token `<end_of_scale>` is inserted after each scale image to indicate the end of different scales.

## 4.2 Training

**Stage 1: Multi-modal Generative Pre-training**. In this stage, the model is initially trained on 500 million image-text pairs to develop comprehensive multimodal representations. Of these training samples, 300 million are directly sampled from web-scale datasets (*i.e.* Laion-2B [55], Coyo-700M [7], Wukong [25] and SA-1B [31]) while the remaining 200 million consist of images from these datasets paired with captions synthesized by existing MLLMs (*i.e.* InternVL-8B [15]). During this process, the textual parameters of the model remain frozen, with only the newly-added vision-specific parameters (*i.e.*, the visual encoder, MLP projector, and MoE visual experts) being trainable.

To enhance the alignment between visual and textual features in more complex multimodal contexts, the model is subsequently trained on 185 million high-quality data consisting of both multimodal

Table 1: **Comparison with existing MLLMs on general MLLM benchmarks.** "#A-Param" denotes the number of activated parameters. [†]InternVL-2.5-2B adopts the same LLM and high-quality data with NaViL, so we mark it as the compositional counterpart. Note that its 300M visual encoder is distilled from another 6B large encoder. **Bold** and underline indicate the best and the second-best performance among native MLLMs, respectively. * denotes our reproduced results. For MME, we sum the perception and cognition scores. Average scores are computed by normalizing each metric to a range between 0 and 100.

| Model | #A-Param | Avg | MMVet | MMMU | MMB | MME | MathVista | OCRBench | CCB |
|---|---|---|---|---|---|---|---|---|---|
| *Compositional MLLMs:* | | | | | | | | | |
| MobileVLM-V2-1.7B [16] | 1.7B | – | – | – | 57.7 | – | – | – | – |
| MobileVLM-V2-3B [16] | 3.0B | – | – | – | 63.2 | – | – | – | – |
| Mini-Gemini-2B [34] | 3.5B | – | 31.1 | 31.7 | 59.8 | 1653 | 29.4 | – | – |
| MM1-3B-MoE-Chat [48] | 3.5B | – | 42.2 | 38.6 | 70.8 | 1772 | 32.6 | – | – |
| DeepSeek-VL-1.3B [40] | 2.0B | 42.3 | 34.8 | 32.2 | 64.6 | 1532 | 31.1 | 409 | 37.6 |
| PaliGemma-3B [6] | 2.9B | 45.6 | 33.1 | 34.9 | 71.0 | 1686 | 28.7 | 614 | 29.6 |
| MiniCPM-V-2 [66] | 2.8B | 51.1 | 41.0 | 38.2 | 69.1 | 1809 | 38.7 | 605 | 45.3 |
| InternVL-1.5-2B [14] | 2.2B | 54.7 | 39.3 | 34.6 | 70.9 | 1902 | 41.1 | 654 | 63.5 |
| Qwen2VL-2B [63] | 2.1B | 58.6 | 49.5 | 41.1 | 74.9 | 1872 | 43.0 | 809 | 53.7 |
| [†]InternVL-2.5-2B [13] | 2.2B | 67.0 | 60.8 | 43.6 | 74.7 | 2138 | 51.3 | 804 | 81.7 |
| *Native MLLMs:* | | | | | | | | | |
| Fuyu-8B (HD) [5] | 8B | – | 21.4 | – | 10.7 | – | – | – | – |
| SOLO [11] | 7B | – | – | – | – | 1260 | 34.4 | – | – |
| Chameleon-7B[1] [9] | 7B | 13.9 | 8.3 | 25.4 | 31.1 | 170 | 22.3 | 7 | 3.5 |
| EVE-7B [19] | 7B | 33.0 | 25.6 | 32.3 | 49.5 | 1483 | 25.2 | 327 | 12.4 |
| EVE-7B (HD) [19] | 7B | 37.0 | 25.7 | 32.6 | 52.3 | 1628 | 34.2 | 398 | 16.3 |
| Emu3 [65] | 8B | – | 37.2 | 31.6 | 58.5 | – | – | 687 | – |
| VoRA [62] | 7B | – | 33.7 | 32.2 | 64.2 | 1674 | – | – | – |
| VoRA-AnyRes [62] | 7B | – | 33.7 | 32.0 | 61.3 | 1655 | – | – | – |
| EVEv2 [20] | 7B | 53.2 | 45.0 | 39.3 | 66.3 | 1709 | **60.0*** | 702 | 30.8* |
| SAIL [32] | 7B | 53.7 | 46.3 | 38.6* | 70.1 | 1719 | 57.0 | 783 | 24.3* |
| Mono-InternVL [43] | 1.8B | 56.4 | 40.1 | 33.7 | 65.5 | **1875** | 45.7 | 767 | 66.3 |
| NaViL-2B (ours) | 2.4B | **67.1** | **78.3** | **41.8** | 71.2 | 1822 | 50.0 | **796** | **83.9** |

alignment samples and pure language data. In this phase, the textual parameters within the self-attention layers are also unfrozen, enabling more refined cross-modal integration.

**Stage 2: Supervised Fine-tuning.** Following common practice in developing MLLM, an additional supervised fine-tuning stage is adopted. In this stage, all parameters are unfrozen and trained using a relatively smaller (*i.e.* 68 million) but higher quality multimodal dataset.

# 5 Experiment

## 5.1 Experimental Setups

**Evaluation Benchmarks**. We evaluate NaViL and existing MLLMs on a broad range of multimodal benchmarks. Specifically, MLLM benchmarks encompass MMVet [69], MMMU val [70], MMBench-EN test [37], MME [22], MathVista MINI [41], OCRBench [39], and CCBench [37]. Visual question answering benchmarks include TextVQA val [57], ScienceQA-IMG test [42], GQA test dev [27], DocVQA test [47], AI2D test [30], ChartQA test [45], and InfographicVQA test [46]. These benchmarks cover various domains, such as optical character recognition (OCR), chart and document understanding, multi-image understanding, real-world comprehension, *etc*.

**Implementation Details**. By default, NaViL-2B is implemented upon InternLM2-1.8B [59], using its weights as initialization for the text part parameters. The text tokenizer and conversation format are also the same. The total number of parameters is 4.2B, of which the number of activation parameters is 2.4B (including 0.6B of visual encoder). The input images are first padded to ensure its length and width are multiples of 32. The stride of Patch Embedding layer is set to 16. The visual encoder adopts bidirectional attention and 2D-RoPE to capture global spatial relationships, while the LLM adopts causal attention and 1D-RoPE to better inherit its capabilities. In the pre-training phase, the global batch size is 7000 for stage 1 and 4614 for stage 2, respectively. The downsampling rate $\tau$ of

Table 2: **Comparison with existing MLLMs on visual question answering benchmarks.**
[†]InternVL-2.5-2B adopts the same LLM and high-quality data with NaViL, so we mark it as the compositional counterpart. Note that its 300M visual encoder is distilled from another 6B large encoder. * denotes our reproduced results. **Bold** and underline indicate the best and the second-best performance among native MLLMs, respectively.

| Model | #A-Param | Avg | TextVQA | SQA-I | GQA | DocVQA | AI2D | ChartQA | InfoVQA |
|---|---|---|---|---|---|---|---|---|---|
| *Compositional MLLMs:* | | | | | | | | | |
| MobileVLM-V2-3B [16] | 3.0B | – | 57.5 | 70.0 | 66.1 | – | – | – | – |
| Mini-Gemini-2B [34] | 3.5B | – | 56.2 | – | – | 34.2 | – | – | – |
| MM1-3B-MoE-Chat [48] | 3.5B | – | 72.9 | 76.1 | – | – | – | – | – |
| DeepSeek-VL-1.3B [40] | 2.0B | – | 57.8 | – | – | – | 51.5 | – | – |
| PaliGemma-3B [6] | 2.9B | – | 68.1 | – | – | – | 68.3 | – | – |
| MiniCPM-V-2 [66] | 2.8B | – | 74.1 | – | – | 71.9 | 62.9 | – | – |
| InternVL-1.5-2B [14] | 2.2B | 71.7 | 70.5 | 84.9 | 61.6 | 85.0 | 69.8 | 74.8 | 55.4 |
| Qwen2VL-2B [63] | 2.1B | 73.1 | 79.7 | 78.2* | 60.3* | 90.1 | 74.7 | 73.5 | 65.5 |
| [†]InternVL-2.5-2B [13] | 2.2B | 76.5 | 74.3 | 96.2 | 61.2 | 88.7 | 74.9 | 79.2 | 60.9 |
| *Native MLLMs:* | | | | | | | | | |
| Fuyu-8B (HD) [5] | 8B | – | – | – | – | – | 64.5 | – | – |
| SOLO [11] | 7B | – | – | 73.3 | – | – | 61.4 | – | – |
| Chameleon-7B[1] [9] | 7B | 17.9 | 4.8 | 47.2 | – | 1.5 | 46.0 | 2.9 | 5.0 |
| EVE-7B [19] | 7B | 40.8 | 51.9 | 63.0 | 60.8 | 22.0 | 48.5 | 19.5 | 20.0 |
| EVE-7B (HD) [19] | 7B | 54.6 | 56.8 | 64.9 | 62.6 | 53.0 | 61.0 | 59.1 | 25.0 |
| Emu3 [65] | 8B | 67.6 | 64.7 | 89.2 | 60.3 | 76.3 | 70.0 | 68.6 | 43.8 |
| VoRA [62] | 7B | – | 56.3 | 75.9 | – | – | 65.6 | – | – |
| VoRA-AnyRes [62] | 7B | – | 58.7 | 72.0 | – | – | 61.1 | – | – |
| EVEv2 [20] | 7B | 71.7 | 71.1 | **96.2** | 62.9 | 77.4* | 74.8 | 73.9 | 45.8* |
| SAIL [32] | 7B | 71.5 | **77.1** | 93.3 | 58.0* | 78.4* | **76.7** | 69.7* | 47.3* |
| Mono-InternVL [43] | 1.8B | 70.1 | 72.6 | 93.6 | 59.5 | 80.0 | 68.6 | 73.7 | 43.0 |
| NaViL-2B (ours) | 2.4B | **75.1** | 76.9 | 95.0 | 59.8 | **85.4** | 74.6 | **78.0** | **56.0** |

visual multi-scale packing is set to $\sqrt{2}/2$. To demonstrate the scaling capability of our approach, we also trained NaViL-9B based on Qwen3-8B [60]. More details are given in the appendix.

## 5.2 Main Results

In Tab. 1, we compare the performance of our model with existing MLLMs across 7 multimodal benchmarks. Compared to native MLLMs, compositional MLLMs demonstrate superior overall performance. For example, InternVL-2.5-2B outperforms existing native MLLMs on most MLLM benchmarks. This indicates that current native MLLMs still have significant room for performance improvement. In contrast, our proposed NaViL achieves overall performance exceeding all existing native MLLMs with a relatively small paramter size. Compared to the compositional baseline model InternVL-2.5-2B that uses the same LLM, NaViL also achieves comparable performance on most benchmarks. It is worth noting that the 300M visual encoder used by InternVL-2.5-2B is distilled from another pre-trained encoder InternViT-6B [15] with a significantly larger parameter size. This demonstrates the superiority of our visual design methods and visual parameter scaling strategies.

In Tab. 2, we further compare the performance of our model with existing MLLMs on mainstream visual question answering tasks. NaViL's average performance still leads previous state-of-the-art native MLLMs and is roughly on par with compositional baselines that require pre-trained encoders. Specifically, in tests such as DocVQA [49], ChartQA [45] and InfoVQA [46], NaViL significantly outperforms the previous state-of-the-art native MLLM, demonstrating the superiority of using an optimal size visual encoder in processing high-resolution images. However, NaViL's performance still has some gap compared to the best compositional MLLMs. We believe that higher-quality instruction data and more powerful LLMs will further narrow this gap.

---

[1]The performance of Chameleon-7B is from [43].

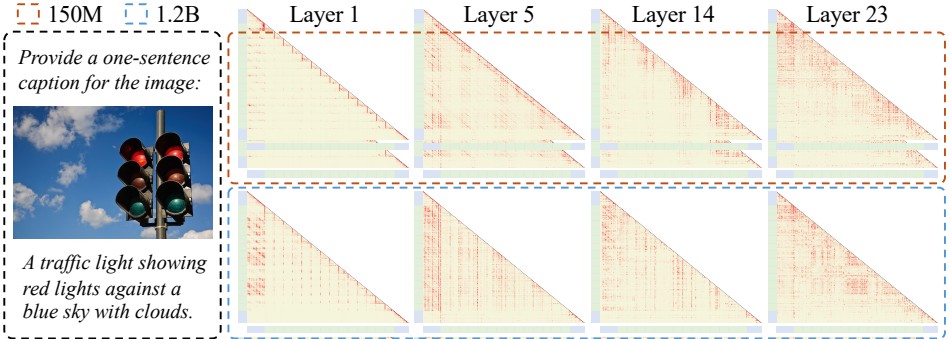

Figure 9: **Visualization of attention maps in LLM-1.8B with different encoder sizes (*i.e*. 150M and 1.2B)**. Text and image tokens are in blue and green, respectively. Larger encoder allows LLMs to attend to global patterns at shallow layers while maintaining higher attention to textual tokens.

## 5.3 Qualitative Experiments

To further analyze the characteristics of native MLLM, we visualized the attention maps of different LLM layers when using encoders of 150M and 1.2B sizes, as shown in Fig. 9. Two findings can be drawn from the figure. First, similar to previous native-MLLMs [43], despite having an encoder, the attention patterns in shallow layers still exhibit obvious locality, gradually shifting toward global information as the depth increases. For example, when using a 150M encoder, image tokens in the first layer tend to attend to spatially adjacent tokens. However, we observe that when the visual encoder is scaled up to 1.2B, visual tokens in shallow layers already begin to attend more to global information. This indicates that a sufficiently large visual encoder can better pre-extract high-level semantic information from the entire image.

Secondly, from a cross-modal interaction perspective, a larger visual encoder also facilitates earlier interaction between visual and language features. When using a 1.2B visual encoder, the attention weights between visual tokens and text tokens in the first layer are significantly higher than those in the 150M counterpart. Earlier interaction is more beneficial for feature alignment between modalities, thus providing an explanatory perspective for the improved performance achieved when using larger encoder sizes. We believe these findings will provide beneficial insights for developing native MLLMs. More visualizations can be found in the supplementary materials.

## 6 Conclusion

This paper systematically investigates native end-to-end training for MLLMs, examining its design space and scaling properties under data constraints. Our study reveals three key insights: 1) Initialization with pre-trained LLMs, combined with visual encoders and MoE architecture, significantly improves performance; 2) Visual encoder scaling is limited by the LLM's capacity, unlike traditional LLM scaling; 3) The optimal encoder size scales log-proportionally with the LLM size. Based on these findings, we propose NaViL, a native MLLM that achieves competitive performance on diverse multimodal benchmarks, outperforming existing compositional MLLMs. We hope these insights will inspire future research on next-generation MLLMs.

**Limitations and Broader Impacts**. Due to limited computation resources, this paper only investigates the scaling properties of native MLLMs up to 9B parameters. Subsequent experiments with larger scales (*e.g*., 30 billion, 70 billion, 100 billion, *etc*.) can be conducted to further validate this scaling trend. In addition, this paper focuses only on visual and linguistic modalities. Future research may explore broader modalities and provide more in-depth insights beyond the current visual-linguistic paradigm.

**Acknowledgments**    The work is supported by the National Key R&D Program of China (NO. 2022ZD0161300, and NO. 2022ZD0160102), by the National Natural Science Foundation of China (U24A20325, 62321005, 62376134), and by the China Postdoctoral Science Foundation (No. BX20250384).

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

# Technical Appendices and Supplementary Material

## A    NaViL-9B: Scaling up to 9B parameters

To further demonstrate the scaling capability of our method, we trained NaViL-9B based on Qwen3-8B [60]. The total number of activation parameters is 9.2B, of which 1.2B belongs to the visual encoder. The training recipe is similar to NaViL-2B, as shown in Tab. 8, except the visual multi-scaling packing is disabled in the first sub-stage of pre-training for acceleration.

Tab. 3 presents a comparison of the total training tokens required by our method versus two compositional counterparts. Notably, our approach achieves comparable performance while using substantially fewer training tokens, demonstrating improved training efficiency.

Table 3: **Comparison between NaViL and existing MLLMs on the number of training tokens.**

| Models | Train ViT | Train MLLM | Total |
|---|---|---|---|
| Qwen2.5VL [4] | unknown | 4.1T | >4.1T |
| InternVL2.5-8B [12] | >3.3T | 140B | >3.5T |
| NaViL-2B (ours) | 0 | 800B | 800B |
| NaViL-9B (ours) | 0 | 450B[1] | 450B |

The performance results on multimodal and visual question answering benchmarks are shown in Tab. 4. With a similar parameter size, our NaViL-9B outperforms all existing native MLLMs by a large margin on almost all benchmarks. Besides that, compared to the compositional baseline model InternVL-2.5-8B with a similar parameter size, NaViL-9B also achieves competitive performance. Such results show that our proposed native MLLM can be scaled up to larger parameter sizes and achieve consistent performance gains.

## B    More discussions on Compositional MLLMs and Native MLLMs

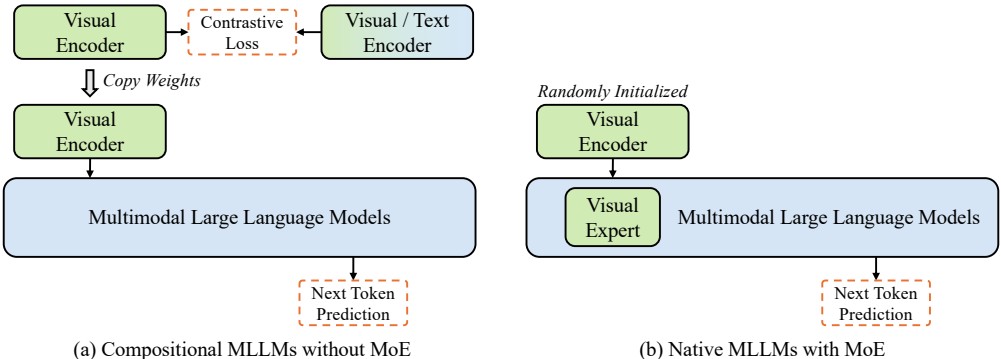

(a) Compositional MLLMs without MoE    (b) Native MLLMs with MoE

Figure 10:    **Paradigm Comparison between Compositional MLLMs and Native MLLMs.** Compositional MLLMs adopt different training objectives and strategies (*e.g.* Contrastive Loss or Next-Token-Prediction) to pre-train the visual encoder and LLM separately, while native MLLMs optimize both image and text components in an end-to-end manner using a unified training objective (*i.e.* Next-Token-Prediction).

Fig. 10 further illustrates the difference between compositional MLLMs and native MLLMs. Compositional MLLMs typically have different components initialized by separate unimodal pre-training, where different training objectives and strategies are employed to train the LLM and visual encoder. For example, the visual encoder can be trained using an image-text contrastive learning objective (e.g., CLIP [52], SigLIP [72]) or a self-supervised learning objective (e.g., DINOv2 [51]). The complexity

---

[1]Due to limited computational resource and time, current version of NaViL-9B in this paper is only trained with 450B tokens.

of such training process increases the difficulty of scalability. On the other hand, as discussed in [56], native MLLM optimizes both image and text modalities end-to-end using a unified training objective (i.e., next-token prediction (NTP)). This avoids introducing additional bias and significantly simplifies the scaling effort.

## C  More Related Works

**Research on Neural Scaling Laws.**   The foundational work on Neural Scaling Laws began in the Natural Language Processing (NLP) domain, where [29] established predictable power-law relationships demonstrating that performance loss ($L$) scales reliably with model size ($N$) and data size ($D$), and that larger, decoder-only Transformer models are more compute-efficient. Following works [23] further extended such research to encoder-decoder architectures, observing consistency in scaling exponents on Neural Machine Translation (NMT) tasks. Driven by these successes, in the vision domain, [71] confirmed the applicability of scaling laws to Vision Transformers (ViT), systematically demonstrating continuous performance improvement by scaling both model size (up to 2 billion parameters) and training data. Most recently, these principles have been generalized to Large Multimodal Models, where [1] developed scaling laws that unify the contributions of text, image, and speech modalities by explicitly modeling synergy and competition as an additive term. Furthering this, [56] explored Native Multimodal Models (NMMs) using Mixture of Experts (MoEs), finding an unbalanced scaling law that suggests scaling training tokens ($D$) is more critical than scaling active parameters ($N$) as the compute budget grows.

## D  Implementation Details

The hyperparameters of model architecture for NaViL-2B and NaViL-9B are listed in Tab. 6, while the hyperparameters of training recipe for NaViL-2B and NaViL-9B are provided in Tab. 7 and Tab. 8, respectively. The high-quality multimodal data used in Pre-training and Supervised Fine-tuning is from InternVL-2.5 [12], which is sourced from various domains, such as image captioning, general question answering, multi-turn dialogue, charts, OCR, documents, and knowledge, *etc*.; while the pure language data is primarily from InternLM2.5 [8].

## E  The NLP capability

We also evaluate the NLP capability of our model on three popular NLP tasks, as shown in Tab. 5. Thanks to the modality-specific MoE architecture, NaViL maintains the NLP capabilities of its initialization LLM (Qwen3-8B). Despite not using a large amount of high-quality text data, NaViL performs well on the common NLP tasks and show much stronger NLP capabilities compared to other native MLLMs, showing its data efficiency.

## F  More Qualitative Results

More visualization results of multimodal understanding are provided below.

---

[2]The performance of Chameleon-7B is from [43].

Table 4: **Comparison between NaViL-9B and existing MLLMs on multimodal benchmarks.** "#A-Param" denotes the number of activated parameters. [†]InternVL-2.5-8B adopts the same high-quality data with NaViL-9B, so we mark it as the compositional counterpart. Note that its 300M visual encoder is distilled from another 6B large encoder. * denotes our reproduced results. **Bold** and underline indicate the best and the second-best performance among native MLLMs, respectively. For MME, we sum the perception and cognition scores. Average scores are computed by normalizing each metric to a range between 0 and 100.

| Model | #A-Param | Avg | MMVet | MMMU | MMB | MME | MathVista | OCR-B | TVQA | DocVQA | AI2D | ChartQA | InfoVQA |
|---|---|---|---|---|---|---|---|---|---|---|---|---|---|
| *Compositional MLLMs:* | | | | | | | | | | | | | |
| MobileVLM-V2 [16] | 1.7B | — | — | — | 57.7 | — | — | — | — | — | — | — | — |
| MobileVLM-V2 [16] | 3.0B | — | — | — | 63.2 | — | — | — | 57.5 | — | — | — | — |
| Mini-Gemini [34] | 3.5B | — | 31.1 | 31.7 | 59.8 | 1653 | 29.4 | — | 56.2 | 34.2 | — | — | — |
| MM1-MoE-Chat [48] | 3.5B | — | 42.2 | 38.6 | 70.8 | 1772 | 32.6 | — | 72.9 | — | — | — | — |
| DeepSeek-VL [40] | 2.0B | — | 34.8 | 32.2 | 64.6 | 1532 | 31.1 | 409 | 57.8 | — | 51.5 | — | — |
| PaliGemma [6] | 2.9B | — | 33.1 | 34.9 | 71.0 | 1686 | 28.7 | 614 | 68.1 | — | 68.3 | — | — |
| MiniCPM-V-2 [66] | 2.8B | — | 41.0 | 38.2 | 69.1 | 1809 | 38.7 | 605 | 74.1 | 71.9 | 62.9 | — | — |
| InternVL-1.5 [14] | 2.2B | 61.3 | 39.3 | 34.6 | 70.9 | 1902 | 41.1 | 654 | 70.5 | 85.0 | 69.8 | 74.8 | 55.4 |
| Qwen2VL [63] | 2.1B | 67.3 | 49.5 | 41.1 | 74.9 | 1872 | 43.0 | 809 | 79.7 | 90.1 | 74.7 | 73.5 | 65.5 |
| InternVL-2.5 [13] | 2.2B | 69.6 | 60.8 | 43.6 | 74.7 | 2138 | 51.3 | 804 | 74.3 | 88.7 | 74.9 | 79.2 | 60.9 |
| Qwen2VL [63] | 8.2B | 77.1 | 62.0 | 54.1 | 83.0 | 2327 | 58.2 | 866 | 84.3 | 94.5 | 83.0 | 83.0 | 76.5 |
| Qwen2.5-VL [4] | 8.2B | 80.2 | 67.1 | 58.6 | 83.5 | 2347 | 68.2 | 864 | 84.9 | 95.7 | 83.9 | 87.3 | 82.6 |
| [†]InternVL-2.5 [13] | 8.1B | 77.3 | 62.8 | 56.0 | 84.6 | 2344 | 64.4 | 822 | 79.1 | 91.9 | 84.5 | 84.8 | 75.7 |
| *Native MLLMs:* | | | | | | | | | | | | | |
| Fuyu-8B (HD) [5] | 8B | — | 21.4 | — | 10.7 | — | — | — | — | — | 64.5 | — | — |
| SOLO [11] | 7B | — | — | — | — | 1260 | 34.4 | — | — | — | 61.4 | — | — |
| Chameleon-7B[2] [9] | 7B | 14.0 | 8.3 | 25.4 | 31.1 | 170 | 22.3 | 7 | 4.8 | 1.5 | 46.0 | 2.9 | 5.0 |
| EVE-7B [19] | 7B | 34.6 | 25.6 | 32.3 | 49.5 | 1483 | 25.2 | 327 | 51.9 | 22.0 | 48.5 | 19.5 | 20.0 |
| EVE-7B (HD) [19] | 7B | 45.2 | 25.7 | 32.6 | 52.3 | 1628 | 34.2 | 398 | 56.8 | 53.0 | 61.0 | 59.1 | 25.0 |
| Emu3 [65] | 8B | — | 37.2 | 31.6 | 58.5 | — | — | 687 | 64.7 | 76.3 | 70.0 | 68.6 | 43.8 |
| VoRA [62] | 7B | — | 33.7 | 32.2 | 64.2 | 1674 | — | — | 56.3 | — | 65.6 | — | — |
| VoRA-AnyRes [62] | 7B | — | 33.7 | 32.0 | 61.3 | 1655 | — | — | 58.7 | — | 61.1 | — | — |
| EVEv2 [20] | 7B | 62.3 | 45.0 | 39.3 | 66.3 | 1709 | 60.0* | 702 | 71.1 | 77.4* | 74.8 | 73.9 | 45.8* |
| SAIL [32] | 7B | 63.7 | 46.3 | 38.6* | 70.1 | 1719 | 57.0 | 783 | 77.1 | 78.4* | 76.7 | 69.7* | 47.3* |
| Mono-InternVL [43] | 1.8B | 60.6 | 40.1 | 33.7 | 65.5 | 1875 | 45.7 | 767 | 72.6 | 80.0 | 68.6 | 73.7 | 43.0 |
| NaViL-2B (ours) | 2.4B | 68.8 | 78.3 | 41.8 | 71.2 | 1822 | 50.0 | 796 | 76.9 | 85.4 | 74.6 | 78.0 | 56.0 |
| NaViL-9B (ours) | 9.2B | **77.0** | **79.6** | **54.7** | **76.5** | **2225** | **66.7** | **837** | **77.2** | **90.6** | **82.4** | **85.4** | **70.2** |

Table 5: **Comparison of NaViL and existing native MLLMs on three common NLP tasks.** Except for Chameleon, models are evaluated using OpenCompass toolkit [18].

| Models | #A-Param | MMLU | CMMLU | MATH |
|---|---|---|---|---|
| InternLM2-Chat [59] | 1.8B | 47.1 | 46.1 | 13.9 |
| Qwen3-8B (non-thinking) [60] | 8B | 76.5 | 76.8 | 71.1 |
| EVE [19] | 7B | 43.9 | 33.4 | 0.7 |
| Chameleon [9] | 7B | 52.1 | - | 11.5 |
| Mono-InternVL [43] | 2B | 45.1 | 44.0 | 12.3 |
| NaViL-9B (ours) | 9.2B | 74.9 | 75.1 | 66.2 |

Table 6: **Hyper-Parameters of Model Architecture.**

| Component | Hyper-Parameter | NaViL-2B | NaViL-9B |
|---|---|---|---|
| visual encoder | # Params | 0.6B | 1.2B |
| | depth | 24 | 32 |
| | width | 1472 | 1792 |
| | MLP width | 5888 | 7168 |
| | # attention heads | 23 | 28 |
| LLM (w/ MoE) | # experts | 2 | 2 |
| | # A-Params | 1.8B | 8.0B |
| | depth | 24 | 36 |
| | width | 2048 | 4096 |
| | MLP width | 8192 | 12288 |
| | # attention heads | 16 | 32 |

Table 7: **Hyper-parameters for training NaViL-2B.**

| Configuration | Multi-modal Generative Pre-training (S1) | | Supervised Fine-tuning (S2) |
|---|---|---|---|
| | S1.1 | S1.2 | |
| Maximum number of image patches | 4096 | 12188 | 24576 |
| Training steps | 70k | 40k | 30k |
| Global batch size | $7,000$ | $4,614$ | $2,234$ |
| Weight decay | 0.05 | 0.1 | 0.01 |
| Learning rate schedule | | constant with warm-up | cosine decay |
| Peak learning rate | | $5e^{-5}$ | $2e^{-5}$ |
| Visual Multi-scale Packing | | ✓ | |
| LLM max sequence length | | $16,384$ | |
| Warm-up steps | | 200 | |
| Optimizer | | AdamW | |
| Optimizer hyperparameters | | $\beta_1 = 0.9, \beta_2 = 0.95, eps = 1e^{-8}$ | |
| Gradient accumulation | | 1 | |
| Numerical precision | | `bfloat16` | |

Table 8: **Hyper-parameters for training NaViL-9B.**

| Configuration | Multi-modal Generative Pre-training (S1) | | Supervised Fine-tuning (S2) |
|---|---|---|---|
| | S1.1 | S1.2 | |
| Maximum number of image patches | 4096 | 12188 | 24576 |
| Training steps | 50k | 33k | 6k |
| Weight decay | 0.05 | 0.1 | 0.01 |
| Global batch size | $10,300$ | $1,792$ | $3,520$ |
| Visual Multi-scale Packing | ✗ | ✓ | ✓ |
| Learning rate schedule | | constant with warm-up | cosine decay |
| Peak learning rate | | $5e^{-5}$ | $2e^{-5}$ |
| LLM max sequence length | | $16,384$ | |
| Warm-up steps | | 200 | |
| Optimizer | | AdamW | |
| Optimizer hyperparameters | | $\beta_1 = 0.9, \beta_2 = 0.95, eps = 1e^{-8}$ | |
| Gradient accumulation | | 1 | |
| Numerical precision | | `bfloat16` | |

**Image Captioning and Visual Question Answering**

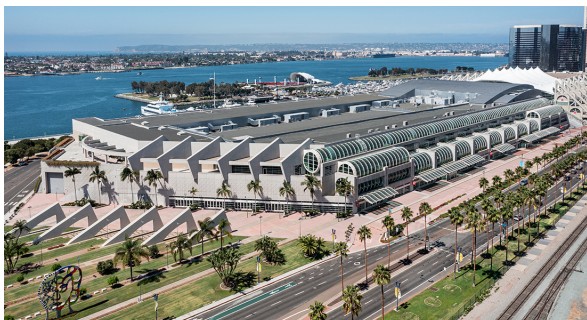

**Prompt:** Describe this image in detail.

**Response:** The image shows an aerial view of a modern architectural complex situated near a coastal area. The building has a distinctive design with a series of curved, wave-like structures on its roof, giving it a futuristic appearance. The exterior is primarily composed of light-colored materials, possibly concrete or stone, with large glass windows.

In front of the building, there is a wide, multi-lane road lined with palm trees, suggesting a warm, coastal climate. The road appears to be a major thoroughfare, with designated bike lanes and pedestrian pathways. There are also train tracks running parallel to the road, indicating the presence of a public transportation system.

The background features a large body of water, likely a bay or ocean, with several boats and yachts docked at a marina. The coastline is lined with more buildings, and there are hills or mountains visible in the distance, adding to the scenic view. The overall setting suggests a bustling urban area with a focus on modern infrastructure and a strong connection to the waterfront.

**Prompt:** Which city is this building most likely in?

**Response:** This building is most likely in San Diego, California. The architectural style and the surrounding landscape, including the water and palm trees, are characteristic of San Diego.

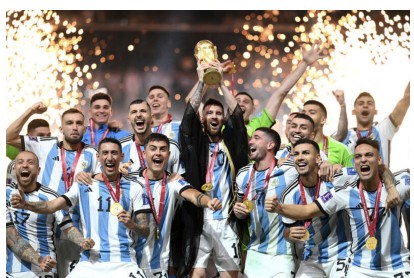

**Prompt:** Who is the person in black in the middle?

**Response:** The person in black in the middle is Lionel Messi.

**Prompt:** What are they celebrating about?

**Response:** The image shows a group of soccer players celebrating with a trophy, wearing medals and jerseys. They are likely celebrating a victory in a soccer tournament or championship.

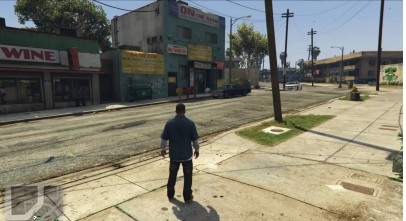

**Prompt:** What is the content in this image?

**Response:** The image appears to be a screenshot from a video game, likely Grand Theft Auto V (GTA V), given the visual style and interface elements. It shows a character standing on a sidewalk in a small town or suburban area. The environment includes a street with tram tracks, a few buildings with signs, and some cars parked along the road. The buildings have signs for businesses like "Wine" and "On The Bone." The scene has a somewhat deserted look, with no other characters or significant activity visible.

**OCR**

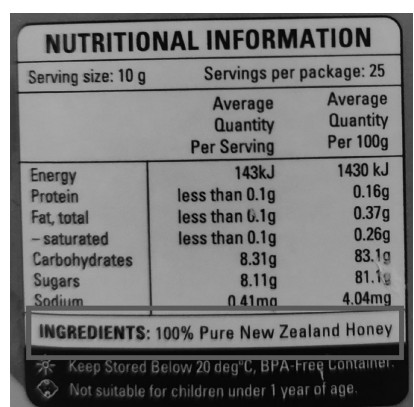

**Prompt:** Extract all the text in this image.

**Response:** NUTRITIONAL INFORMATION
Serving size: 10 g
Servings per package: 25
Average Quantity Per Serving | Average Quantity Per 100g
Energy | 143kJ | 1430 kJ
Protein | less than 0.1g | 0.16g
Fat, total | less than 0.1g | 0.37g
- saturated | less than 0.1g | 0.26g
Carbohydrates | 8.31g | 83.1g
Sugars | 8.11g | 81.1g
Sodium | 0.41mg | 4.04mg
INGREDIENTS: 100% Pure New Zealand Honey
Keep Stored Below 20 deg°C, BPA-Free Container.
Not suitable for children under 1 year of age.

---

**Prompt:** Describe the image.

**Response:** The image is a receipt from Chapelbank Hotel for a lunch order for five people. The items listed on the receipt include 3x soup at £3.95 each, totaling £11.85; lunch main course at £6.95; lasagne (V) at £6.95; misc food at £6.95; and 2x chips/onion rings at £1.75 each, totaling £3.50. The subtotal for the food is £29.25. The receipt also includes a previous balance of £29.25, making the total amount due £29.25. The hotel's address is 69 East High Street, Forfar DD8 2EP, and the telephone number is 01307 463151.

**Prompt:** How many people are attending this lunch?

**Response:** There are 5 people attending this lunch. This is indicated at the top of the receipt where it says "Food = Lunch x 5 people".

---

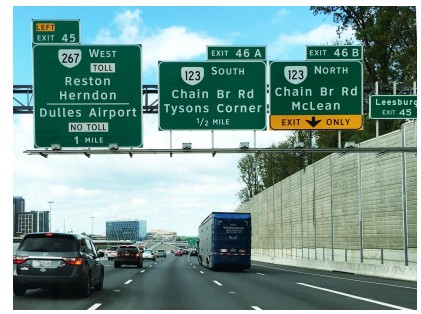

**Prompt:** Where is the left exit going to?

**Response:** The left exit is going to Reston, Herndon, and Dulles Airport.

**Prompt:** Which road leads to the EXIT 46 B?

**Response:** The road that leads to EXIT 46 B is Chain Br Rd North.

**Multi-lingual OCR**

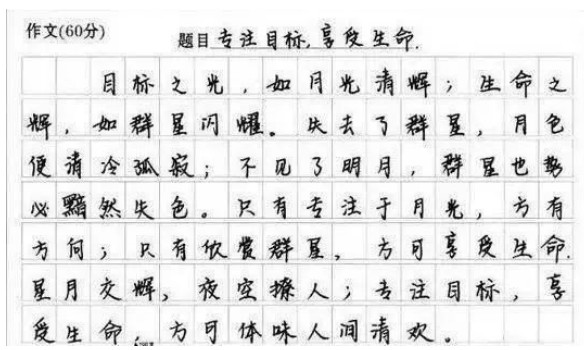

**Prompt:** 请先提取里面的文本，再翻译为英文。

**Response:**
文本内容:
作文（60分） 题目：专注目标，享受生命。
目标之光，如月光清辉；生命之辉，如群星闪耀。失去了群星，月色便显得冷寂；不见了明月，群星也势必黯然失色。只有专注于月光，方有方向；只有欣赏群星，方可享受生命。星月交辉，夜空撩人；专注目标，享受生命，方可体味人间清欢。
翻译为英文:
Essay (60 points)
Title: Focus on your goal, enjoy life.
The light of the goal is like the clear glow of moonlight; the brilliance of life is like the dazzling stars. Without the stars, the moonlight becomes cold and desolate; without the moon, the stars will inevitably become dull. Only by focusing on the moonlight can you find direction; only by appreciating the stars can you enjoy life. The stars and moon shine together, making the night sky enchanting; focusing on your goal, enjoying life, allows you to savor the quiet joy of life.

**Document Understanding**

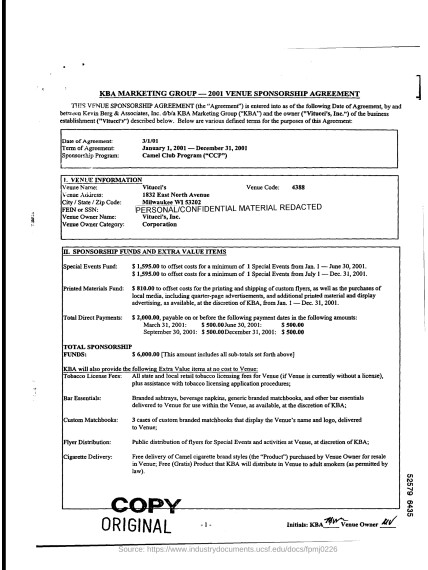

**Prompt:** What is the duration of this venue sponsorship agreement?

**Response:** January 1, 2001 - December 31, 2001

**Prompt:** Where is Vitucci's located (complete address)?

**Response:** 1832 East North Avenue, Milwaukee WI 53202

**Prompt:** What cigarette brand is mentioned in the agreement for delivery and distribution?

**Response:** Camel cigarette brand styles purchased by Venue Owner for resale in Venue

**Prompt:** Who has discretion over the distribution of flyers for special events?

**Response:** KBA Marketing Group ("KBA") has discretion over the distribution of flyers for special events.

**LaTeX and Math**

$$\mathcal{J}_{GRPO}(\theta) = \mathbb{E}[q \sim P(Q), \{o_i\}_{i=1}^{G} \sim \pi_{\theta_{old}}(O|q)]$$

$$\frac{1}{G} \sum_{i=1}^{G} \frac{1}{|o_i|} \sum_{t=1}^{|o_i|} \left\{ \min \left[ \frac{\pi_\theta(o_{i,t}|q, o_{i,<t})}{\pi_{\theta_{old}}(o_{i,t}|q, o_{i,<t})} \hat{A}_{i,t}, \text{clip}\left( \frac{\pi_\theta(o_{i,t}|q, o_{i,<t})}{\pi_{\theta_{old}}(o_{i,t}|q, o_{i,<t})}, 1-\varepsilon, 1+\varepsilon \right) \hat{A}_{i,t} \right] - \beta \mathbb{D}_{KL} \left[ \pi_\theta || \pi_{ref} \right] \right\}, \quad (3)$$

**Prompt:** Convert the formula in the figure to latex format.

**Response:**

```latex
\begin{equation*}
    \begin{split}
    \mathcal{J}_{\text{GRPO}}(\theta) &= \mathbb{E}[q \sim P(Q), \{o_i\}_{i=1}^G
    \sim \pi_{\theta_{old}}(O|q)] \\
    & \frac{1}{G} \sum_{i=1}^G \frac{1}{|o_i|} \sum_{t=1}^{|o_i|} \left\{ \min
    \left[ \frac{\pi_{\theta}(o_{i,t}|q, o_{i,<t})}{\pi_{\theta_{old}}(o_{i,t}|q, o_{i,<t})}
    \hat{A}_{i,t}, \text{clip}
    \left( \frac{\pi_{\theta}(o_{i,t}|q, o_{i,<t})}{\pi_{\theta_{old}}(o_{i,t}|q, o_{i,<t})},
    1 - \varepsilon, 1 + \varepsilon \right) \hat{A}_{i,t} \right] -
    \beta \mathcal{D}_{KL} \left[ \pi_{\theta} || \pi_{ref} \right] \right\},
    \end{split}
\end{equation*}
```

---

**Prompt:** Find $x$. $A = 148 \ m^2$. Please think step by step.

**Response:** To find the value of $x$ in the parallelogram, we can use the formula for the area of a parallelogram:

$$A = \text{base} \times \text{height}$$

Given: - Area $A = 148 \, \text{m}^2$ - Base $= 8 \, \text{m}$ - Height $= x \, \text{m}$
Substitute the given values into the formula:

$$148 = 8 \times x$$

Solve for $x$:

$$x = \frac{148}{8}$$
$$x = 18.5 \, \text{m}$$

So, $x = 18.5 \, \text{m}$.

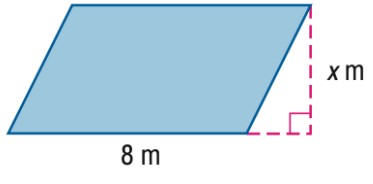

8 m, x m

---

**Prompt:** A dog show enthusiast recorded the weight of the winning dog at recent dog shows. According to the table, what was the rate of change between 2016 and 2017? (Unit: kilograms per year)

| Weight of winning dog at a dog show | |
|---|---|
| Year | Weight (kg) |
| 2013 | 27 |
| 2014 | 28 |
| 2015 | 34 |
| 2016 | 29 |
| 2017 | 40 |

**Response:** To determine the rate of change between 2016 and 2017, we use the formula for rate of change:

$$\text{Rate of Change} = \frac{\text{Weight in 2017} - \text{Weight in 2016}}{\text{Year 2017} - \text{Year 2016}}$$

From the table: - Weight in 2016 = 29 kg - Weight in 2017 = 40 kg
Substitute these values into the formula:

$$\text{Rate of Change} = \frac{40 \, \text{kg} - 29 \, \text{kg}}{2017 - 2016} = \frac{11 \, \text{kg}}{1 \, \text{year}}$$

$$\text{Rate of Change} = 11 \, \text{kg/year}$$

Thus, the rate of change between 2016 and 2017 is $\boxed{11 \, \text{kg/year}}$.

