# OpenReview forum: "NaViL: Rethinking Scaling Properties of Native Multimodal Large Language Models under Data Constraints"
_NeurIPS.cc/2025/Conference — NeurIPS 2025 poster_

### Official Review · Reviewer_KNVW · 2025-06-30

**Clarity:** 2
**Significance:** 3
**Originality:** 2
**Rating:** 5
**Confidence:** 4

**Summary:**

* This paper presents a systematic study on how to best design and scale “native” Multimodal Large Language Models (MLLMs), which are trained end-to-end.
* The authors empirically investigate several key design choices under a “data constraint”: the LLM initialization, the use of *modality-specific* MoEs, and visual encoder architecture.
* The primary contribution is the derivation of a scaling law for these native MLLMs: the optimal visual encoder size is limited by the LLM's capacity and should scale log-proportionally with the LLM's size.
* Based on these principles, the authors propose NaViL, a model that achieves competitive performance on 14 benchmarks, often with significantly higher training efficiency than “compositional” counterparts.

**Questions:**

The core scientific contribution of this paper is strong, but several issues hold back its impact in the presentation. Addressing the following points would greatly improve the paper:

1. **Architectural Diagram:** As noted in W1, the paper is difficult to understand without an architectural diagram. To address this, could you please add a figure that clearly illustrates the NaViL architecture? It would be very helpful to see the data flow from the trainable vision encoder through the connector to the LLM, and to visualize the differences between the MoE and non-MoE variants you tested.
2. **Clarification of Terminology:** To resolve the ambiguity mentioned in W2, could you more precisely define "native" vs “compositional” MLLMs?
   1. “Early-fusion” a la Chameleon had a more clear-cut definition. What is the defining characteristic of a “native” model? That the *vision model* is trained from scratch?
   2. What if the vision model were initialized with pretrained vision-model weights (just as the paper advocates for initializing the LLM component with pretrained LLM weights)—would this still be a *native* MLLM?
3. **Minor Presentation Issues:** Please address the minor issues listed in W3.

**Ethical Concerns:**

["NO or VERY MINOR ethics concerns only"]

**Final Justification:**

all major concerns addressed. I think this is a good paper worthy of acceptance

**Limitations:**

No.

These scaling laws are validated only at a small scale, though there are “final results” in the appendix up to 9B.

It is unrealistic to run these experiments, so perhaps a limitation should be listed that it is unknown if the observed scaling trends would hold at drastically larger scale (\~30B, \~70B, 100+B, etc).... ie, caveat the observed scaling laws by emphasizing they are observed at a small scale.

**Quality:**

4

**Strengths And Weaknesses:**

**Strengths:**

1. The paper's core strength is its systematic and well-designed empirical study in Section 3\. The experiments are clean, the visualizations are clear and informative (e.g., Figs. 2–6), and the derived "Observations" provide valuable design principles for the community.
2. The main finding—that the optimal visual encoder size should scale with the LLM size—provides a clear, data-driven “recipe” that challenges the common practice of using fixed-size vision encoders in compositional MLLMs.
3. The resulting NaViL models demonstrate impressive performance, outperforming prior native MLLMs and competing with SOTA compositional models of comparable scale. The supplement highlights the model's training efficiency, achieving these results with a fraction of the training tokens used by other large models.

**Weaknesses:**

1. **Clarity / Self-Containedness:** There is a lack of clarity regarding the model's architecture. The paper is not well self-contained in this regard, and requires the reader to reference external work (e.g., Mono-InternVL) to understand crucial architectural details. A clear architectural diagram that clearly displays the exact model setup (and possibly contrasts it with the “compositional” counterpart) would be a significant improvement.
2. **Vague Terminology:** The terms “native” and “compositional” MLLMs are not clearly defined or well-contrasted with existing community terms like "early-fusion" (most standard) or "monolithic".
   1. A diagram could help clear this up additionally (see W1)
3. **Minor Writing and Presentation Issues:**
   1. The paper has several minor but frequent English writing errors (e.g., "with Data Constraint," "Besides,") that reduce its overall polish. Sending it through a grammar/spell checker would help.
   2. Some mathematical notation is overloaded (e.g., the symbol $\\tau$ is used for two different concepts in §3.3.2 and §4.1), which could be easily rectified.
   3. Citation \[24\] referencing pretraining vision encoders on L20 seems like a mis-citation? This citation appears to be about weight interpolation

---

> ### Author Rebuttal · Authors · 2025-07-31
>
> Thank you for your valuable and detailed comments. We address your questions as follows.
>
> > **W1: Clarity / Self-Containedness: There is a lack of clarity regarding the model's architecture. The paper is not well self-contained in this regard, and requires the reader to reference external work (e.g., Mono-InternVL) to understand crucial architectural details. A clear architectural diagram that clearly displays the exact model setup (and possibly contrasts it with the “compositional” counterpart) would be a significant improvement.**
>
> A1: The model architecture is introduced in text form in the paper. L95-103 first defines the meta architecture of MLLM, consisting of a visual encoder (with an MLP connector), an LLM, and a MoE architecture injected into the LLM. L262-270 further describes the detailed architecture of NaViL-2B, including model parameter size, attention mechanism, positional embeddings, and more. Appendix C explains the detailed implementation of the modality-specific MoE adopted by our approach.
>
> Due to the policy of NeurIPS 2025, we were unable to provide the architectural diagram during the rebuttal process. However, to further clarify the model configuration, the following table lists the key hyperparameters of NaViL-2B. Both the table and the figure will be updated in the final version. The code for the model structure and inference will also be open-sourced.
>
> | Component | Hyper-Parameter | Value |
> |------|------|------|
> | **visual encoder** | # Params | 0.6B |
> | | depth | 24 |
> | | width | 1472 |
> | | MLP width | 5888 |
> | | attention heads | 23 |
> | **LLM (w/ MoE)** | # experts | 2 |
> | | # A-Params | 1.8B |
> | | depth | 24 |
> | | width | 2048 |
> | | MLP width | 8192 |
> | | attention heads | 16 |
>
> > **W2: Vague Terminology: The terms “native” and “compositional” MLLMs are not clearly defined or well-contrasted with existing community terms like "early-fusion" (most standard) or "monolithic".**
> >
> > **A diagram could help clear this up additionally (see W1)**
>
> A2: Please refer to A7. In addition, a figure for comparing “native” and “compositional” MLLMs will be included in the final version.
>
> > **W3.1: The paper has several minor but frequent English writing errors (e.g., "with Data Constraint," "Besides,") that reduce its overall polish. Sending it through a grammar/spell checker would help.**
>
> A3: Thanks for the suggestion. The paper will be polished to eliminate such errors in the final version.
>
> > **W3.2: Some mathematical notation is overloaded (e.g., the symbol  is used for two different concepts in §3.3.2 and §4.1), which could be easily rectified.**
>
> A4: Thanks for the suggestion. Different notations will be used in the final version.
>
> > **W3.3: Citation [24] referencing pretraining vision encoders on L20 seems like a mis-citation? This citation appears to be about weight interpolation**
>
> A5: Thanks for the suggestion. However, citation [24] is not a mis-citation, as it refers to the repository page for the OpenCLIP software project on Zenodo (one of the pre-trained visual encoders widely adopted by MLLMs), not to weight interpolation.
>
> > **Q1: Architectural Diagram: As noted in W1, the paper is difficult to understand without an architectural diagram. To address this, could you please add a figure that clearly illustrates the NaViL architecture? It would be very helpful to see the data flow from the trainable vision encoder through the connector to the LLM, and to visualize the differences between the MoE and non-MoE variants you tested.**
>
> A6: Please refer to A1. An additional figure will be included in the final version, illustrating both the data flow and the differences between MoE and non-MoE variants.
>
> > **Q2: Clarification of Terminology: To resolve the ambiguity mentioned in W2, could you more precisely define "native" vs “compositional” MLLMs?**
> >
> > **1. “Early-fusion” a la Chameleon had a more clear-cut definition. What is the defining characteristic of a “native” model? That the vision model is trained from scratch?**
> >
> > **2. What if the vision model were initialized with pretrained vision-model weights (just as the paper advocates for initializing the LLM component with pretrained LLM weights)—would this still be a native MLLM?**
>
> A7: (1) The distinction between native MLLMs and compositional MLLMs primarily lies in whether the training process and training objectives can be unified.
>
> As discussed in L69-71, compositional MLLMs have different components initialized by separate unimodal pre-training, where different training objectives and strategies are employed to train the LLM and visual encoder. For example, the visual encoder can be trained using an image-text contrastive learning objective (e.g., CLIP, SigLIP) or a self-supervised learning objective (e.g., DINOv2). The complexity of such training process increases the difficulty of scalability.
>
> As shown in L79-83, the generally accepted definition of a native MLLM [1] is that it optimizes both image and text modalities end-to-end using a unified training objective (i.e., next-token prediction (NTP)). This avoids introducing additional bias and significantly simplify the scaling effort.
>
> In contrast, **early-fusion (and monolithic) models** define MLLMs from a model architecture perspective, meaning that visual and text modality inputs are fused in the early stage and no modality-specific parameters are used. For example, Chameleon is both an early-fusion model and a native MLLM because it integrates visual and textual inputs from the outset and trains the entire model end-to-end using a unified training objective.
>
> (2) Initializing the vision encoder with a contrastive learnining pre-trained model introduces different training objectives and thus violates the definition of native MLLM in our paper.
>
> > **Q3: These scaling laws are validated only at a small scale, though there are “final results” in the appendix up to 9B.
> It is unrealistic to run these experiments, so perhaps a limitation should be listed that it is unknown if the observed scaling trends would hold at drastically larger scale (~30B, ~70B, 100+B, etc).... ie, caveat the observed scaling laws by emphasizing they are observed at a small scale.**
>
> A8: Thanks for the suggestion. We will mention this as our limitation and potential future work in the final version.
>
> [1] Shukor, Mustafa, Enrico Fini, Victor Guilherme Turrisi da Costa, Matthieu Cord, Joshua Susskind, and Alaaeldin El-Nouby. "Scaling laws for native multimodal models." arXiv preprint arXiv:2504.07951. 2025.

---

> ### Comment · Reviewer_KNVW · 2025-08-05
>
> thanks for the detailed rebuttal. most of my concerns have been addressed.
>
> **A1:**
> I indeed read the lines you reference in your submission but found them confusing and insufficient to understand. this was the point of writing this weakness.
>
> please be sure to include a good architecture diagram (A1, A6) in the final version.
>
> **A7:**
> this explanation was very helpful, thank you. a high-level conceptual diagram or much better explanation in the paper would   be a big improvement here.

---

> > ### Author Response · Authors · 2025-08-05
> >
> > Thank you for your thoughtful feedback. We are delighted to hear that most of your concerns have been addressed through our rebuttal. As committed, we will include comprehensive architectural diagrams (A1, A6) and a high-level conceptual diagram (A7) in the final version to further enhance the paper's clarity.

---

### Official Review · Reviewer_Y9RL · 2025-07-02

**Clarity:** 4
**Significance:** 4
**Originality:** 4
**Rating:** 6
**Confidence:** 5

**Summary:**

This paper presents a systematic study on training native Multimodal Large Language Models (MLLMs) in an end-to-end manner, with particular focus on design space exploration and scaling properties under data constraints. The authors derive several valuable insights, including: (1) the necessity of initializing from pretrained LLMs, (2) the benefits of adopting Mixture-of-Experts (MoE) architecture, (3) comprehensive analysis of visual encoder design choices, and (4) the finding that scaling visual encoders logarithmically proportional to LLMs outperforms fixed-size visual encoders. The extensive experimental validation leads to the development of NaViL, a native MLLM that achieves competitive performance across various multimodal benchmarks while outperforming existing compositional MLLMs. The paper is well-written and provides clear experimental insights.

**Questions:**

The related work section should include more comprehensive discussion of visual scaling paradigms, particularly foundational work such as Zhai et al. (2022) "Scaling Vision Transformers" (CVPR), which established important principles for visual model scaling that are relevant to this work.

**Ethical Concerns:**

["NO or VERY MINOR ethics concerns only"]

**Final Justification:**

I appreciate the author's response, which truly addressed all my questions. I highly appreciate the author's 7B model in the appendix, which demonstrates the generalization of the scaling property. I believe this paper will bring new insights to the field, so I decided to increase my score.

**Limitations:**

Yes

**Quality:**

4

**Strengths And Weaknesses:**

## Strengths

1. The motivation is compelling and addresses a practically important problem. Investigating the scaling properties of native MLLMs represents a valuable research direction that could significantly simplify the training pipeline for building robust MLLMs.

2. The analytical experiments are extensive and yield actionable insights for the community. The investigation follows a logical progression: first providing detailed quantitative analyses of individual design choices like  LLM initialization strategies and MoE architecture adoption. This systematic approach effectively demonstrates the scaling characteristics of native MLLMs.

3. The experimental results are convincing and demonstrate clear advantages.   NaViL  achieves competitive performance compared to compositional counterparts while requiring significantly less multimodal training data. The preserved NLP capabilities further validate the approach's effectiveness compared to previous native MLLM methods.

4. The paper is well-structured and accessible. The introduction clearly establishes motivation, the analysis proceeds logically, and comprehensive figures and tables support the findings effectively.

## Weaknesses

1. The final result of the article only provides a 2B-sized model. It is better to conduct experiments on a larger model to verify the effectiveness of the scaling law.

2. The paper lacks detailed discussion of computational overhead and training efficiency comparisons between native and compositional approaches, which is crucial for practical adoption.

---

> ### Author Rebuttal · Authors · 2025-07-31
>
> Thank you for your valuable and detailed comments. We address your questions as follows.
>
> > **W1: The final result of the article only provides a 2B-sized model. It is better to conduct experiments on a larger model to verify the effectiveness of the scaling law.**
>
> A1: We also report the results of NaViL-9B in Appendix A. With a similar parameter size, our NaViL-9B outperforms all existing native MLLMs by a large margin on almost all benchmarks. Besides that, compared to the compositional baseline model InternVL-2.5-8B with a similar parameter size, NaViL-9B also achieves competitive performance.  Such results demonstrate that NaViL can be scaled up to larger parameter sizes and achieve consistent performance gains.
>
> > **W2: The paper lacks detailed discussion of computational overhead and training efficiency comparisons between native and compositional approaches, which is crucial for practical adoption.**
>
> A2: As described in L143-146 and L262-271, NaViL adopts a modality-specific MoE architecture, resulting in the same number of parameters and inference FLOPs as the compositional counterpart. The implementation details of the modality-specific MoE are provided in Appendix C. A comparison of total training tokens is shown in Table 1 in the Appendix. Our approach achieves comparable performance while using significantly fewer training tokens, resulting in improved training efficiency.
>
> > **Q1: The related work section should include more comprehensive discussion of visual scaling paradigms, particularly foundational work such as Zhai et al. (2022) "Scaling Vision Transformers" (CVPR), which established important principles for visual model scaling that are relevant to this work.**
>
> A3: Thank you for your suggestion. More discussion on the visual scaling paradigm will be included in the related work section of the final version. “Scaling Vision Transformers” (Zhai et al., 2022) is an important and valuable work related to our research, which will also be discussed in the final version.

---

> > ### Comment · Reviewer_Y9RL · 2025-08-05
> > **Reply to Authors' Rebuttal**
> >
> > I appreciate the author's response, which truly addressed all my questions. I highly appreciate the author's 7B model in the appendix, which demonstrates the generalization of the scaling property. I believe this paper will bring new insights to the field, so I decided to increase my score.

---

> > > ### Author Response · Authors · 2025-08-05
> > >
> > > We sincerely appreciate your valuable suggestions and are pleased that our response has addressed all your questions. Your recognition of our 7B model results and the generalization of the scaling property is highly valued. We are grateful for your confidence in our work's contribution to the field and honored by your decision to increase the score.

---

### Official Review · Reviewer_kjaE · 2025-07-05

**Clarity:** 3
**Significance:** 2
**Originality:** 2
**Rating:** 4
**Confidence:** 4

**Summary:**

This paper investigates the design space and scaling properties of native multimodal large language models (MLLMs) under data-constrained conditions. The authors distinguish between former compositional approaches, which separately pretrained vision encoders and language models before combining them, and native approaches that jointly optimize both components end-to-end. Through experimentation across three architectural components, the authors propose NaViL, a native MLLM that achieves competitive performance with approximately 600 million pre-training image-text pairs. The key claimed contribution is the discovery that optimal visual encoder size scales log-linearly with language model size.

**Questions:**

1.	Section 3.2.2 lacks crucial implementation details about the MoE architecture. What is the exact MoE configuration used in the comparison? How many experts are employed, and what is the expert selection mechanism? Most critically, what are the precise parameter counts for both vanilla transformer and MoE configurations to ensure fair comparison? Without parameter equivalence, the observed performance improvements in Figure 3 may simply reflect increased model capacity rather than architectural advantages.
2.	Section 3.2.3 explores visual encoder architecture only through width-depth variations. Have the authors considered more sophisticated architectural explorations such as attention mechanism variations, different normalization strategies, or alternative activation functions that might yield more novel insights for native MLLM design?
3.	While the log-linear relationship in Figure 7 is empirically observed, what theoretical principles could explain this phenomenon? Besides, the only three pairs (0.5B, 2B, 7B) of datapoints are quite limited for such a claim of scaling law. Can the authors make sure this trend can generalize to larger scales?
4.	Baseline Evaluation. Tables 1 and 2 show incomplete results for baselines evaluation across multiple benchmarks. Would complete baseline evaluation change the relative performance rankings?
5.	As mentioned in Section 5.1, NaViL-2B is built upon InternLM2-1.8B, initializing its text part parameters with the weights of InternLM2-1.8B. I assume this means that NaViL-2B is, to some extent, further trained based on InternLM2-1.8B. While this approach may facilitate training convergence, is it really fair for the performance comparisons? If NaViL-2B were trained from scratch, would it still achieve similar performance results?

**Ethical Concerns:**

["NO or VERY MINOR ethics concerns only"]

**Final Justification:**

The rebuttal has addressed most of my concerns.

**Limitations:**

yes

**Paper Formatting Concerns:**

This paper does not exhibit any noticeable formatting issues.

**Quality:**

2

**Strengths And Weaknesses:**

Strengths
1.	This paper is very well written and easy to understand.
2.	The research question that this paper wants to explore is significant. The native MLLM training approach should be more explored and compared against the compositional approaches.
3.	The log-linear scaling relationship between optimal visual encoder size and language model size represents a potentially significant discovery. This finding challenges compositional paradigms that employ fixed visual encoder sizes across varying LLM scales and could provide quantitative guidance for future model design.
4.	The research addresses real-world constraints by demonstrating competitive performance using approximately 600 million pre-training image-text pairs.
Weaknesses
1.	Section 3.2 presents limited novel insights across its three subsections. The LLM initialization exploration (Section 3.2.1) yields unsurprising conclusions that initialization of pre-trained weights help. The visual encoder architecture study (Section 3.2.3) only examines width-depth trade-offs and reaches conventional conclusions that shallow models converge faster while deeper models achieve better performance, consistent with established deep learning principles.
2.	Section 3.2.2 (lines 137-151) provides inadequate description of the MoE architecture implementation and fails to ensure fair experimental comparison. The paper does not specify the exact parameter counts for vanilla versus MoE configurations, the number of experts used, or the expert selection mechanism.
3.	The downsampling rate mentioned in line 232 is represented by the symbol \tau, which is also used to represent another quantity used in line 211, potentially causing ambiguity.
4.	Tables 1 and 2 contain substantial missing experimental results across multiple baseline models and benchmarks and without further explanation. This incomplete evaluation results undermines the paper's claims about competitive performance against existing methods and makes it difficult to assess the true relative standing of NaViL.

---

> ### Author Rebuttal · Authors · 2025-07-31
>
> Thank you for your valuable and detailed comments. We address your questions as follows.
>
> > **W1.1: Section 3.2 presents limited novel insights across its three subsections. The LLM initialization exploration (Section 3.2.1) yields unsurprising conclusions that initialization of pre-trained weights help.**
>
> A1: We understand your concerns about LLM initialization benefits seeming intuitive. However, this intuition doesn't always hold even in computer vision; for example, [1] shows ImageNet pretraining doesn't improve object detection and instance segmentation performance.
>
> In the MLLM community, while LLM initialization has become the default practice, this fundamental question remains unanswered. Therefore, we quantitatively demonstrate LLM initialization's impact in Fig. 2, breaking this black box and addressing this fundamental question for the entire community. From this perspective, our conclusions and results provide valuable insights to the community.
>
> > **W1.2: (1) Section 3.2.3 only examines width-depth trade-offs and reaches conventional conclusions that shallow models converge faster while deeper models achieve better performance. (2) Have the authors considered more sophisticated architectural explorations such as attention mechanism variations and different normalization strategies?**
>
> A2: (1) Similar to W1.1, visual encoder width-depth design for MLLMs remains controversial. Previous studies [2,3] focus on single visual encoders for pure vision tasks, but these principles may not transfer to MLLMs with large-scale LLMs. Most works use fixed-size visual encoders (e.g., SigLIP) or discard them entirely (Mono-InternVL), making it difficult to draw clear conclusions about width-depth design for native MLLMs.
>
> We systematically ablated visual encoder width-depth design and found **a different conclusion** that visual encoders achieve near-optimal performance across wide depth/width ranges. Except for extremely shallow/deep configurations, relatively shallow encoders converge faster while deeper ones perform better in later training stages. These findings provide valuable guidance for future native MLLM research.
>
> (2) We focus on encoder size scaling properties, keeping meta-architecture (attention, normalization, activation functions) the same while varying only depth and width, following common practices for scaling law research [4,7]. Better architectural components will be explored in future work.
>
> > **W2: Section 3.2.2 (lines 137-151) provides inadequate description of the MoE architecture implementation and fails to ensure fair experimental comparison. The paper does not specify the exact parameter counts for vanilla versus MoE configurations, the number of experts used, or the expert selection mechanism.**
>
> A3: The number of activated experts is set to one (L145-L146) and static modality routing is used (L143-145). Therefore, **the activation parameter count and training FLOPs are the same for both vanilla and MoE configurations**, which ensures fair comparison and is consistent with the experimental configurations in many prior works [4].
> Appendix C has introduced the details of MoE implementation. We thank the suggestion and will revise by discussing more implementation details of our MoE in the main paper.
>
> > **W3: The downsampling rate mentioned in line 232 is represented by the symbol \tau, which is also used to represent another quantity used in line 211, potentially causing ambiguity.**
>
> A4: Thanks for the suggestion. Different symbols will be used in L232 and L211 for the camera-ready version to avoid ambiguity.
>
> > **W4: Tables 1 and 2 contain substantial missing experimental results across multiple baseline models and benchmarks and without further explanation. This incomplete evaluation results undermines the paper's claims about competitive performance against existing methods and makes it difficult to assess the true relative standing of NaViL.**
>
> A5: In Tab. 1 and 2, we report only the official experimental results of other baseline models, sourced from their respective papers. Since different papers evaluate their models on different benchmarks, some results are missing. To further support our claims about NaViL's performance, we report the full results of Qwen2VL-2B, SAIL, and EVEv2, as shown in the following tables (* denotes our reproduced results).
>
> | Model | #A-Param | Avg | Avg (w/o CCB) | MMVet | MMMU | MMB | MME | MathVista | OCRBench | CCB |
> |-------|----------|-----|---------------|-------|------|-----|-----|-----------|----------|-----|
> | EVEv2 | 7B | 53.2 | 57 | 45 | 39.3 | 66.3 | 1709 | 60* | 702 | 30.8* |
> | SAIL | 7B | 53.7 | 58.6 | 46.3 | 38.6* | 70.1 | 1719 | 57 | 783 | 24.3* |
> | NaViL-2B | 2.4B | 67.1 | 64.3 | 78.3 | 41.8 | 71.2 | 1822 | 50 | 796 | 83.9 |
> | NaViL-9B | 9.2B | 73.2 | 73.4 | 79.6 | 54.7 | 76.5 | 2225 | 66.7 | 837 | 71.6 |
>
>
> | Model | #A-Param | Avg | TextVQA | SQA-I | GQA | DocVQA | AI2D | ChartQA | InfoVQA |
> |-------|----------|-----|---------|-------|-----|--------|------|---------|---------|
> | Qwen2VL-2B | 2.1B | 73.1 | 79.7 | 78.2* | 60.3* | 79.7 | 74.7 | 73.5 | 65.5 |
> | EVEv2 | 7B | 71.7 | 71.1 | 96.2 | 62.9 | 77.4* | 74.8 | 73.9 | 45.8* |
> | SAIL | 7B | 71.5 | 77.1 | 93.3 | 58.0* | 78.4* | 76.7 | 69.7* | 47.3* |
> | NaViL-2B | 2.4B | 75.1 | 76.9 | 95 | 59.8 | 85.4 | 74.6 | 78 | 56 |
> | NaViL-9B | 9.2B | 80.4 | 76.4 | 98.1 | 59.5 | 90.6 | 82.4 | 85.4 | 70.2 |
>
> The conclusion remains consistent: NaViL's average performance is still ahead of previous native MLLMs and is competitive with the compositional baselines.
>
> > **Q1: Section 3.2.2 lacks crucial implementation details about the MoE architecture. What is the exact MoE configuration used in the comparison? How many experts are employed, and what is the expert selection mechanism? Most critically, what are the precise parameter counts for both vanilla transformer and MoE configurations to ensure fair comparison? Without parameter equivalence, the observed performance improvements in Figure 3 may simply reflect increased model capacity rather than architectural advantages.**
>
> A6: Please refer to A3.
>
> > **Q2: Section 3.2.3 explores visual encoder architecture only through width-depth variations. Have the authors considered more sophisticated architectural explorations such as attention mechanism variations, different normalization strategies, or alternative activation functions that might yield more novel insights for native MLLM design?**
>
> A7: Please refer to A2.
>
> > **Q3: While the log-linear relationship in Figure 7 is empirically observed, what theoretical principles could explain this phenomenon? Besides, the only three pairs (0.5B, 2B, 7B) of datapoints are quite limited for such a claim of scaling law. Can the authors make sure this trend can generalize to larger scales?**
>
> A8: Several theories explain the log-linear relationship in Fig. 7, including encoder-decoder scaling laws in neural machine translation [5] and multimodal scaling laws for text, speech, and images [6]. Similar theories could apply to native MLLMs, but our paper focuses on design space and scaling properties under data constraints, leaving theoretical analysis for future work.
>
> Scaling laws predict larger-scale model performance from small-scale experiments. Previous works [7] don't show all datapoints for every scale. Since native MLLM scaling research typically uses models below 8B parameters [4], we follow this convention while planning verification on larger MLLMs. We will further verify the results of the scaling law on a larger MLLM.
>
> > **Q4: Baseline Evaluation. Tables 1 and 2 show incomplete results for baselines evaluation across multiple benchmarks. Would complete baseline evaluation change the relative performance rankings?**
>
> A9: Please refer to A5.
>
> > **Q5: As mentioned in Section 5.1, NaViL-2B is built upon InternLM2-1.8B, initializing its text part parameters with the weights of InternLM2-1.8B. I assume this means that NaViL-2B is, to some extent, further trained based on InternLM2-1.8B. While this approach may facilitate training convergence, is it really fair for the performance comparisons? If NaViL-2B were trained from scratch, would it still achieve similar performance results?**
>
> A10: NaViL-2B's textual parameters are initialized from InternLM2-1.8B. Since this paper focuses on native MLLM scaling properties under data constraints, convergence speed is crucial. All MLLMs in Tab. 1-2 use pre-trained LLM initialization for practical high performance, ensuring fair comparison. As shown in L113-115 and Fig. 2, while non-LLM initialization can achieve similar final performance, it requires substantial high-quality data, violating our data-constrained setting.
>
> [1] He, Kaiming, Ross Girshick, and Piotr Dollár. "Rethinking imagenet pre-training." In ICCV. 2019.
>
> [2] Alabdulmohsin, Ibrahim M., Xiaohua Zhai, Alexander Kolesnikov, and Lucas Beyer. "Getting vit in shape: Scaling laws for compute-optimal model design." In NeurIPS. 2023.
>
> [3] Zhai, Xiaohua, Alexander Kolesnikov, Neil Houlsby, and Lucas Beyer. "Scaling vision transformers." In CVPR. 2022.
>
> [4] Shukor, Mustafa, Enrico Fini, Victor Guilherme Turrisi da Costa, Matthieu Cord, Joshua Susskind, and Alaaeldin El-Nouby. "Scaling laws for native multimodal models." arXiv preprint arXiv:2504.07951. 2025.
>
> [5] Ghorbani, Behrooz, Orhan Firat, Markus Freitag, Ankur Bapna, Maxim Krikun, Xavier Garcia, Ciprian Chelba, and Colin Cherry. "Scaling Laws for Neural Machine Translation." In ICLR. 2022.
>
> [6] Aghajanyan, Armen, Lili Yu, Alexis Conneau, Wei-Ning Hsu, Karen Hambardzumyan, Susan Zhang, Stephen Roller, Naman Goyal, Omer Levy, and Luke Zettlemoyer. "Scaling laws for generative mixed-modal language models." In ICML. 2023.
>
> [7] Kaplan, Jared, Sam McCandlish, Tom Henighan, Tom B. Brown, Benjamin Chess, Rewon Child, Scott Gray, Alec Radford, Jeffrey Wu, and Dario Amodei. "Scaling laws for neural language models." arXiv preprint arXiv:2001.08361. 2020.

---

> > ### Author Response · Authors · 2025-08-07
> >
> > Dear Reviewer kjaE,
> >
> > Thank you for your thoughtful review and constructive feedback. As the discussion period nears its end, we want to ensure our responses have adequately addressed your concerns.
> > We are encouraged that both Reviewer Y9RL and kNVW found our rebuttal helpful, and Reviewer Y9RL has indicated a willingness to raise the score. We sincerely hope that our rebuttal has also resolved the issues you identified. Should any questions remain, we would be grateful for the opportunity to provide further clarification.
> > Thank you again for your valuable time and expertise.
> >
> > Best regards.

---

> > ### Author Response · Authors · 2025-08-08
> >
> > Dear reviewer kjaE,
> >
> > We fully understand your busyness, and also sincerely hope that our efforts would be recognized by you. To date, we have received two strongly positive scores by other reviewers. Thus, we are still looking forward to your new decision based on our response. Thank you!
> >
> > Best regards!

---

> > ### Comment · Reviewer_kjaE · 2025-08-09
> >
> > Thank you for the responses! It has addressed most of my concerns.

---

### Decision · Program_Chairs · 2025-09-17

**Decision:**

Accept (poster)

**Comment:**

This paper presents a systematic study of end-to-end training for native Multimodal Large Language Models (MLLMs) under data constraints.
The authors analyze key design choices, including LLM initialization, modality-specific Mixture-of-Experts (MoE), and visual encoder architectures. A central contribution is the derivation of a scaling law, showing that visual encoders should scale log-proportionally to LLM size rather than remain fixed. Building on these findings, the authors introduce NaViL, a native MLLM that matches or surpasses compositional approaches, which separately pretrained vision encoders and language models before combining them. Extensive experiments validate these insights and demonstrate their impact on multimodal performance and training efficiency.

All reviewers were positive and ask for clarification about implementation details, architecture types, scaling laws explanation, computational overhead and experiments.   The reviewers appreciated the authors' answers. Most of their concerns have been addressed. All remain positive or very positive.  The AC proposes to follow their proposal for acceptance.